# Analysis of Visuo Motor Control between Dominant Hand and Non-Dominant Hand for Effective Human-Robot Collaboration

**DOI:** 10.3390/s20216368

**Published:** 2020-11-08

**Authors:** Hanjin Jo, Woong Choi, Geonhui Lee, Wookhyun Park, Jaehyo Kim

**Affiliations:** 1Department of Mechanical and Control Engineering, Handong Global University, Pohang 37554, Korea; 21834006@handong.edu (H.J.); 21834004@handong.edu (G.L.); 21834002@handong.edu (W.P.); 2Department of Information and Computer Engineering, National Institute of Technology, Gunma College, Maebashi 371–8530, Japan

**Keywords:** visuomotor, hand dominance, control mechanism, spatio-temporal, tracking target

## Abstract

The human-in-the-loop technology requires studies on sensory-motor characteristics of each hand for an effective human–robot collaboration. This study aims to investigate the differences in visuomotor control between the dominant (DH) and non-dominant hands in tracking a target in the three-dimensional space. We compared the circular tracking performances of the hands on the frontal plane of the virtual reality space in terms of radial position error (Δ*R*), phase error (Δ*θ*), acceleration error (Δ*a*), and dimensionless squared jerk (*DSJ*) at four different speeds for 30 subjects. Δ*R* and Δ*θ* significantly differed at relatively high speeds (Δ*R*: 0.5 Hz; Δ*θ*: *0*.5, 0.75 Hz), with maximum values of ≤1% compared to the target trajectory radius. *DSJ* significantly differed only at low speeds (0.125, 0.25 Hz), whereas Δ*a* significantly differed at all speeds. In summary, the feedback-control mechanism of the DH has a wider range of speed control capability and is efficient according to an energy saving model. The central nervous system (CNS) uses different models for the two hands, which react dissimilarly. Despite the precise control of the DH, both hands exhibited dependences on limb kinematic properties at high speeds (0.75 Hz). Thus, the CNS uses a different strategy according to the model for optimal results.

## 1. Introduction

Human–robot collaboration has been emerging with the recent advances in smart manufacturing [1,2,3,4]. The human–robot collaboration is challenging owing to the different mechanisms used by humans and current robots. A robot controls an end effector with a position control mechanism based on absolute coordinates, whereas a human controls limbs with a predictive-control mechanism and feedback-control mechanism based on body coordinates [5,6,7,8]. Humans use a combination of feedback and predictive-control mechanisms. The dominant control mechanism depends on the periodicity of the target [9,10,11,12,13,14,15]. The difference in the mechanism leads to the human control of the upper limb with low precision and accuracy compared to the robot. To achieve an effective collaboration in smart manufacturing, the characteristics of human control mechanisms need to be analyzed for robots’ planning, identification, and classification of the human movement (how people react to spatio-temporal changes of the target).

The human controls the upper limb based on the sensory-motor integration system in body coordinates, which helps humans achieve goals and well adapt to spatio-temporal changes [16,17,18]. The upper-limb control, achieved by a complex muscle coordination, entails a combination of a feedback-control mechanism and predictive-control mechanism in a closed-loop control [19,20,21,22,23,24,25,26,27,28]. Tracking movements, representative of visuomotor control, is a fundamental example of these mechanisms in action. Studies on neural motor control during tracking movements focused on mechanisms of human control of their limbs toward a target and typically concluded that it involves feedback- and predictive-control mechanisms. The feedback-control mechanism in the visual tracking movement of the upper limb consists of three parts, muscle control, visual feedback information, and afferent feedback information from the upper-limb muscles [24,25]. The motor command signal that activates the muscle is determined by visual feedback information from the central nervous system (CNS), with a time period up to approximately 3 ms related to the refractory period [26,27,28,29]. The afferent feedback information from the upper-limb muscle determines the smoothness and accuracy of the upper-extremity movement [30].

In circular tracking movements, parameters related to position, phase, and velocity were used to identify the movement mechanisms of humans [31,32,33,34,35,36,37,38,39]. For a quantitative upper-limb functional evaluation, the parameters for an analysis should be different according to the dimensions of the target [31,32,33]. In addition, the results of circular tracking movements are strongly related to the target periodicity [31,34,35,36,37,38,39]. If the target has a long cycle, under 0.5 Hz, the position-error corrective mechanism becomes dominant. Alternatively, the predictive-control mechanism becomes dominant [21,36,38]. All these control mechanisms relate to the attribute of intermittency, which was derived from Craik’s observation in 1947 [38,40,41,42,43,44,45,46]. This observation is related to a particular area of the brain, known as the cerebellum. Regarding physiological aspects, the cerebellum is involved in motion planning and control, detects errors, and provides intermittent corrections related to movements [43].

Although numerous studies have been conducted on human control mechanisms in tracking movements, few studies compared them between the dominant hand (DH) and non-dominant hand (NDH). The few studies that compared the DH and NDH investigated either physiological or functional aspects. It was recently reported that left handedness is connected to brain imaging patterns and genotypes [47]. Postnatal handedness can be explained as babies starting to determine their hand preferences in 18 gestational weeks [48]. In terms of functional aspect, the previous studies mainly focused on analyzing differences, such as finger tapping forces and accuracy toward a path or target, between the DH and NDH. They contributed to the understanding of the differences between hands through inverse kinematics of upper-limb movement stemming from the CNS. The previous studies focused on the accuracy, phase error, and effects of other sensory information for analyzing functional differences [49,50,51,52,53] and demonstrated no statistically significant differences with regard to psychomotor skills and hand usage between the DH and NDH during visually guided movements. However, Mathew et al. reported that sensory processing is independent of motor control and that the DH is superior to the NDH in motor skills, except for predictive control [53].

Therefore, we aimed to investigate the differences in visuomotor ability between the DH and NDH in the three-dimensional (3D) space. We investigated differences between visuomotor control outcomes and mechanisms of the DH and NDH according to the target periodicity. Recently, we developed an experimental 3D system for visuomotor control in a virtual reality (VR) environment [54,55]. Thus, we analyzed the visuomotor control characteristics of circular tracking movements between the DH and NDH in the 3D space using four parameters: radial position error (Δ*R*), phase error (Δ*θ*), 3D acceleration error (Δ*a*), and dimensionless squared jerk (*DSJ*). *DSJ* is a derivative of the acceleration function, which can be used for computing movement smoothness, from the initial point to the final point [56,57]. Studies on upper-limb movements using *DSJ* involved mainly the optimization theory and smoothness analysis. The minimum jerk cost function has been utilized for analyzing optimization hypotheses during point-to-point tasks or reaching movements [56,57,58,59,60,61]. *DSJ* has also been used to compute movement smoothness in the voluntary movements of humans and animals [60,61]. Therefore, we used trajectory smoothness as the degree of dominance for feedback control in the 3D circular tracking movement. We also analyzed variations in visuomotor control in 3D target tracking movements at four different target speeds, using these parameters, between the DH and NDH.

## 2. Materials and Methods

### 2.1. Subjects and Experimental Setup

Thirty healthy subjects (21 males and 9 females) participated in the experiments. Their average age was 24.8 years (22–35 years). None of the subjects had participated in similar studies. The DH was evaluated according to the Oldfield handedness inventory before the experiment [62]; 28 subjects were right-handed, whereas two subjects were left-handed. All subjects received sufficient explanation about the experimental procedure and provided written informed consent prior to the experiment. The protocol was approved by the ethics committee of the National Institute of Technology, Gunma College.

We used a quantitative evaluation system for visuomotor control in the 3D VR space to record the movement of each hand at 90 Hz [55]. The system displays the 3D circular tracking movement using the HTC VIVE head-mounted display (HMD). The hand-held controller is a synchronized real-time VIVE tracer. The system can perform visually guided 3D circular tracking tasks with a target and tracer. The target from the HMD is a virtual red ball with a radius of 1.5 cm, while the handle of the controller is a virtual white stick with a length of 20 cm against a black background. The tracer is yellow, has a radius of 1 cm, and is attached to the top of the virtual stick. The tracer position was synchronized with the subject’s hand movements. The controller direction was also synchronized with the virtual stick. The subjects were asked to track the target using a virtual tracer. The target moves along an invisible circular orbit with a radius of 15 cm. Cartesian 3D coordinate data were derived from the position of the controller.

### 2.2. Experimental Task

We performed the experiment using circular tracking movements in the 3D VR frontal plane for each hand to quantify the 3D visuomotor control performance. The subjects sat on a chair designed for the experiment and wore an HMD. The upper body of the subject was fixed for a motion analysis of the upper limb. We ran a calibration to locate the optimal initial target position for the subject’s arm length and height to minimize the effects of different anthropometric parameters on the experimental results, as shown in Figure 1A. To avoid a subject learning effect, the experiment was performed with random counterbalance. 

The target’s circular orbit was symmetrical along a body-centered coordinate system. The target circled along the orbit at 0.125, 0.25, 0.5, and 0.75 Hz. Before the experiment, the subjects repeated the exercise of placing a tracker on a calibrated target three times. As shown in Figure 1, the subjects were then asked to perform 3D circular tracking movements. In total, 48 experimental trials were performed (six trials × four speeds × two hands) by each subject. 

### 2.3. Experimental Task

Cartesian coordinate data (*x*, *y*, *z*) were acquired with the 3D VR system at a sampling rate of 90 Hz. Each graph shown in Figure 1B,C represents a typical example in this experiment. The red line represents the movement of the target, while the black line indicates the movement of the tracer (Figure 1B,C). We calculated the radial position (*R*), phase position (*θ*), acceleration (*a*), and *DSJ*. *R* and *θ* are utilized for analyzing the tracking performance.

Δ*R* is the mean absolute radial position error, i.e., the absolute difference between the tracer’s radial position and target’s radial position,
(1)$$\Delta R=\frac{\;\sum_{k=1}^{n}\left|R_{k}{}_{tracer}-R_{k}{}_{target}\right|}{n}.$$

Δ*θ* is the mean absolute phase error, i.e., the absolute difference between the phase of the tracer and that of the target,
(2)$$\Delta \theta =\frac{\;\sum_{k=1}^{n}\left|\theta_{k}{}_{tracer}-\theta_{k}{}_{target}\right|}{n}.$$

Δ*a* is the mean absolute acceleration error, i.e., the absolute difference between the acceleration of the tracer and that of the target. The acceleration data were filtered with a Butterworth low-pass filter at a cut-off frequency of 7.5 Hz.
(3)$$a=\frac{d^{2}x}{dt^{2}}+\frac{d^{2}y}{dt^{2}}+\frac{d^{2}z}{dt^{2}},$$
(4)$$\Delta a=\frac{\sum_{k=1}^{n}\;\left|a_{k}{}_{tracer}-a_{k}{}_{target}\right|}{n}.$$

*DSJ* was calculated as:(5)$$DSJ=\left(\int_{t_{0}{}^{\circ}}^{t_{180{}^{\circ}}}\left(\frac{d^{3}x}{dt^{3}}+\frac{d^{3}y}{dt^{3}}+\frac{d^{3}z}{dt^{3}}\right)dt\;\right)D^{3}/v_{peak}^{2},$$
where *D* is the total distance between the end effector and final point, and *v_peak_* is the maximum speed during the movement, i.e., the maximum value of v expressed by:(6)$$v=\frac{d^{2}x}{dt^{2}}+\frac{d^{2}y}{dt^{2}}+\frac{d^{2}z}{dt^{2}}.$$

*DSJ*, which is a jerk without dimensions, measures the smoothness of movement. A lower *DSJ* implies a smoother movement path. The initial and final points of *DSJ* are based on each target position at 0° and 180° by means of the acceleration direction change in the *y* axis. The units of Δ*R*, Δ*θ*, and Δ*a* are mm, degree, and mm/s^2^, respectively. *DSJ* is dimensionless. The data visualization was implemented in Matlab (MathWorks, Natick, MA, USA). Figure 2 shows a typical example of data converted according to the speed (*V*_1_ = 0.125 Hz, *V*_2_ = 0.25 Hz, *V*_3_ = 0.5 Hz, *V*_4_ = 0.75 Hz) versus time.

### 2.4. Statistical Analysis

To statistically analyze each parameter, we used a two-way repeated-measure analysis of variance (ANOVA) on hand factors (hand level: DH, NDH) and speed factors (*V*_1_, *V*_2_, *V*_3_, and *V*_4_). The main effect and interaction effect of each parameter were computed using a repeated-measure function in SPSS Statistics V.26 (IBM, Chicago, IL, USA). A posthoc test was performed using pairwise comparisons with Bonferroni correction.

Additionally, we calculated the mean (M), standard error (SE), and standard deviation (SD) of the data. A Mauchly’s sphericity test was performed to validate the ANOVA results. In Appendix A, we considered differences as statistically significant for *p* < 0.05 and highly significant for *p* < 0.01. The effect sizes in Appendix A were estimated using Cohen’s *D*.

## 3. Results

### 3.1. Differences in Control Outcomes between the DH and NDH in the Circular Tracking Movement

We used two parameters (Δ*R* and Δ*θ*) to represent the control error of the human upper extremity during the circular tracking movement (a typical example is shown in Figure 2; Δ*R* for A1 and A2; Δ*θ* for B1 and B2). The control error (Δ*R* and Δ*θ*) tended to increase with the target speed. The two-way repeated-measure ANOVA on the Δ*R* performance differences revealed significant effects of the hand factor to 14.95 ± 4.33 mm. Δ*R* of the NDH was larger than that of the DH at all speeds, except for *V*_1_ (Figure 3B,C). Except for *V*_1_ and *V*_2_ for the DH, the difference in the main effects of the speed factor between the DH and NDH was highly statistically significant (*F*(2.153, 30.148) = 118.052, *p* < 0.001, partial *η*_2_ = 0.803, item B in Appendix A). However, no statistically significant interaction between the hand factor and speed factor was observed (*F*(2.827, 81.993) = 1.843, *p* = 0.149, partial *η*_2_ = 0.060, items C and D in Appendix A).

The two-way repeated-measure ANOVA on the ∆*θ* performance-based differences revealed a significant main effect for the hand (*F*(1, 29) = 27.291, *p* < 0.001, partial *η*_2_ = 0.485, item A in Appendix A). ∆*θ* of the DH increased from 2.10 ± 0.76° at *V*_1_ to 6.72 ± 2.04° at *V*_4_. For the NDH, it changed from 2.09 ± 0.54° at *V*_1_ to 7.95 ± 2.55° at *V*_4_. Thus, Δ*θ* of the NDH was larger than that of the DH at all speeds except for *V*_1_ (Figure 4B,C). A highly statistically significant effect of the hand factor on ∆*θ* was observed at *V*_3_ and *V*_4_ (Figure 4A; item B in Appendix A). The main effect of the speed factor was highly statistically significant (*F*(1.389, 40.281) = 170.424, *p* < 0.001, partial *η*_2_ = 0.855, item B in Appendix A). The interaction between the hand and speed factors was also highly statistically significant (*F*(1.933, 56.050) = 12.718, *p* < 0.001, partial *η*_2_ = 0.305, items C and D in Appendix A). The maximum error of ∆*θ* in the task was 1.231° at the target speed of *V*_4_.

In summary, the DH can exhibit relatively precise movements at high speeds (*V*_3_, *V*_4_), although no difference in control between the DH and NDH was observed at low speeds (*V*_1_, *V*_2_). Additionally, the accuracies of the hands were different only at a particular speed *(V*_3_). These findings indicate that the feedback mechanism of the DH enables reaction to a wider spectrum of target speeds.

### 3.2. Differences in Circular Tracking Movement Based on ∆a and DSJ

We analyzed ∆*a* to indirectly assess the movement of muscles according to the control strategy used by the subjects (typical examples are shown in Figure 2C1,C2). The main effect of the hand factor was statistically significant (*F*(1, 29) = 11.381, *p* = 0.002, partial *η*_2_ = 0.282, item A in Appendix A; Figure 5A). A statistically significant effect of the speed factor was also observed (*F*(1.157, 33.564) = 248.442, *p* < 0.001, partial *η*_2_ = 0.895, item B in Appendix A). The interaction between the hand and speed factors was also statistically significant (*F*(1.615, 46.848) = 3.474, *p* = 0.049, partial *η*_2_ = 0.107, items C and D in Appendix A). The maximum error of ∆*a* in the tasks was 183.011 mm/s^2^ at *V*_3_. ∆*a* of the NDH was higher than that of the DH at all speeds. In addition, ∆*a* tended to increase according to the speed of the target for both hands. 

*DSJ* is utilized to analyze the smoothness of a movement. The two-way repeated-measure ANOVA was used to assess *DSJ* at four speed levels (*V*_1_, *V*_2_, *V*_3_, *V*_4_) for both hands (DH, NDH). The main effect of the hand factor was statistically significant (*F*(1, 29) = 20.868, *p* < 0.001, partial *η*_2_ = 0.418, item A in Appendix A; Figure 6A). *DSJ* tended to decrease with the increase in the target speed, as shown in Figure 6B,C. The main effect of the speed factor was statistically significant (*F*(1.002, 29.047) = 85.913, *p* < 0.001, partial *η*_2_ = 0.748, item B in Appendix A). The interaction between the hand factor and speed factor was also statistically significant (*F*(1.002, 29.047) = 20.156, *p* < 0.001, partial *η*_2_ = 0.410, items C and D in Appendix A). A statistically significant difference between the hands at speeds of 0.125 and 0.25 Hz was observed. The ratios of the mean *DSJ*, representing the *DSJ* ratio of the DH and NDH, were 2.78, 2.58, 1.10, and 1.07 (Figure 6D). The movement of the DH was rougher at relatively low speeds (*V*_1_, *V*_2_), while the smoothness of the hands at relatively high speeds (*V*_3_, *V*_4_) were similar. The feedback-control ability of the DH was superior to that of the NDH at low target speeds (*V*_1_, *V*_2_), while it tended to be similar between the hands at high speeds (*V*_3_, *V*_4_).

## 4. Discussion

We analyzed the visuomotor control characteristics of the 3D circular tracking movement in the VR space. It can be a representative of a steady-state response. Δ*R* and Δ*θ* are the control outcomes derived from the circular tracking movement described in the polar coordinate system. The control outcomes (Δ*R* and Δ*θ*) and Δ*a* were directly proportional to the speed of the target. However, for *DH*, Δ*R* was not statistically significantly different between *V*_1_ and *V*_2_. For *V*_3_, Δ*R* was statistically significantly different between the hands. Therefore, the DH has a wider speed spectrum range in its feedback mechanism capability. In other words, the CNS sustains control of the DH through a feedback-control mechanism at higher target speeds than those for the NDH. 

Overall, the data show that the control outcomes (Δ*R* and Δ*θ*) at low speeds (*V*_1_, *V*_2_) were similar between the DH and NDH, considering the negligible differences in Δ*R* and Δ*θ* between the hands. All values of Δ*a* for the DH were smaller than those for the NDH. Furthermore, all values of Δ*R* for the DH were smaller than those for the NDH except at *V*_1_. The subjects moved their DHs more frequently for correcting errors at low speeds (*V*_1_, *V*_2_) because of the value of *DSJ*. Thus, the DH can be controlled with a smaller amount of total energy as (1) the muscle contributes to the acceleration of joint dynamics [63,64], (2) the upper-limb movement for task accomplishment is the result of muscle coordination [19,20,65], and (3) owing to the energy relationship between the force and distance. The acceleration error is statistically significant for all target speeds, which indicates that the CNS recognizes each hand as a different model and reacts dissimilarly, which can be an evidence of brain lateralization [66]. These implications suggest that humans have an internal model and efficiently control circular tracking [67,68]. Furthermore, we suggest that humans are particularly proficient in controlling their hands. However, these findings are not sufficient to demonstrate that humans can efficiently control all their joints, as only hand data were analyzed.

*DSJ* is related to movement smoothness. A smaller *DSJ* implies a smoother movement [52,53,54,55,56,57,58,59,60,61]. At the target speeds of *V*_1_ and *V*_2_, the degree of smoothness for the NDH was more than twice lower than that of the DH, as shown in Figure 6D. This implies that the DH actively corrects errors from control outcomes, as suggested in previous studies [36,49]. This is directly related to the superior feedback capability of the DH compared to the NDH. It is unlikely that the results are attributed to the predominance of motor learning attributed to the DH because the experiment minimized the number of tasks [69].

The *DSJ* ratio converged to 1 at high target speeds (*V*_3_ and *V*_4_). The movements at these speeds were also smoother. This tendency suggests that the proportion of control attributable to the predictive-control mechanism increased. When the target speed exceeded *V*_3_, the effects on the radial position error and phase error were statistically significant. However, at *V*_4_, only the phase error was statistically significantly different between the hands. Thus, the DH can be controlled more precisely owing to the phase difference. In addition, it is likely that both hands rely on the kinematic characteristics of their limbs at higher target speeds. This dependence of kinematic characteristics can represent the proficiency of the DH in muscle coordination at the high target speed.

The DH performed a circular tracking movement with less total energy and active feedback control. This is attributed to the combination of torques from the wrist, elbow, and shoulder. It is necessary to attach kinetic sensors to the wrist, elbow, and shoulder joint to identify the model (minimum torque change model, minimum jerk cost function model, or another model) used by humans for upper limbs in circular periodic tracking movements [70]. With this model, it would also be necessary to analyze the muscles that are used in circular tracking (antebrachial, biceps brachii, and deltoid muscles). The muscles used for manual tracking are the biceps brachii and deltoid, which are strongly related to typical motions in humans [65]. In addition, the dependence of kinematic characteristics can be demonstrated by utilizing kinetic sensors.

To achieve an effective human-robot collaboration, humans must use a combination of feedback and predictive mechanisms. Both hands react differently according to the target periodicity, which indicates that humans use different combinations of control mechanisms based on the speed of the target. In summary, the results suggest that the end effector of the robot should be controlled under 0.5 Hz or 471 mm/s, at which the human recognizes the target and controls its upper limb with a predictive-control mechanism. By limiting the speed of the end effector, the human-robot collaboration would be more effective, owing to the utilization of the error correction with a feedback mechanism. In addition, the speed value can be the standard setting as a safety factor. The robot can efficiently identify the human movement and control characteristics. These results could be utilized to guide studies on complex hands-over interactions and human-robot information change. We will be able to verify the efficiency of human hands by measuring brain waves with fMRI and joint movements with an inertial measure unit when each hand is used, which will lead to smoother robots for human-robot collaboration. Moreover, the human–robot collaboration can be developed into face-to-face collaboration.

## 5. Conclusions

In the human-in-the-loop technology, studies on the sensory-motor characteristics of each hand are needed for human-robot collaboration with upper limbs. We analyzed the differences in the control mechanism and outcomes between the DH and NDH during circular tracking movements in the 3D VR space. The CNS reacted dissimilarly between the hands. First, the DH had a more precise control than that of the NDH at the highest speed (0.75 Hz). Second, the DH had a wider speed spectrum in its feedback-control mechanism. However, no significant difference in the control outcomes (Δ*R* and Δ*θ*) was observed at low target speeds (0.125, 0.25 Hz). Moreover, a tendency of dependence of both hands on the kinematic characteristics of the upper limb was observed when the speed of the target increased. In summary, the CNS uses different strategies, relying on the model of the particular hand, which produced optimal results in the tracking task. These results could help the robot identify and recognize the human movement, which contributes to a more efficient human cooperation with the robot.

## Figures and Tables

**Figure 1 sensors-20-06368-f001:**
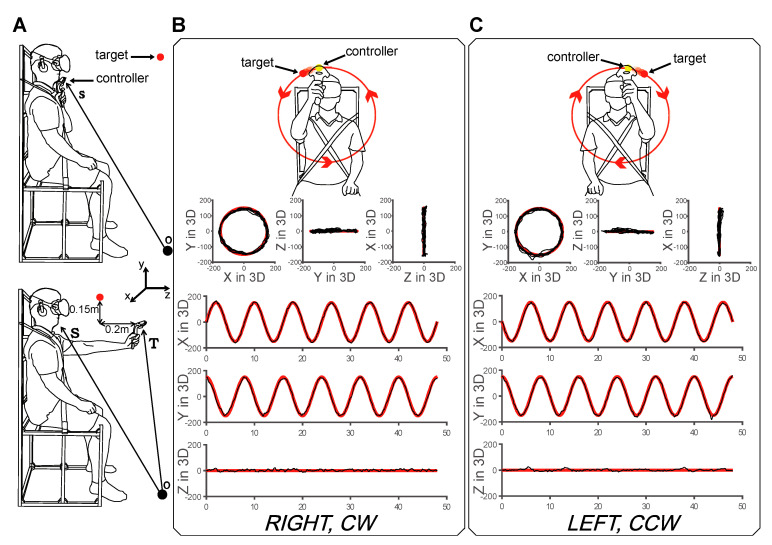
Experimental procedure. (**A**) Calibration of the experiment. The position of the chin of the subject was determined to initialize the position of the *Y* axis of the target. To determine the positions of the target on the *X* and *Z* axes, the system measured the stretched position for the subject’s hand. (**B**) Schematic of the circular tracking movement for the right hand on the frontal VR plane. The target rotated in the clockwise (CW) direction. (**C**) Schematic of the circular tracking movement for the left hand on the frontal plane. The target rotated in the counter-clockwise (CCW) direction. The circular orbit was invisible to the subject throughout the experiment.

**Figure 2 sensors-20-06368-f002:**
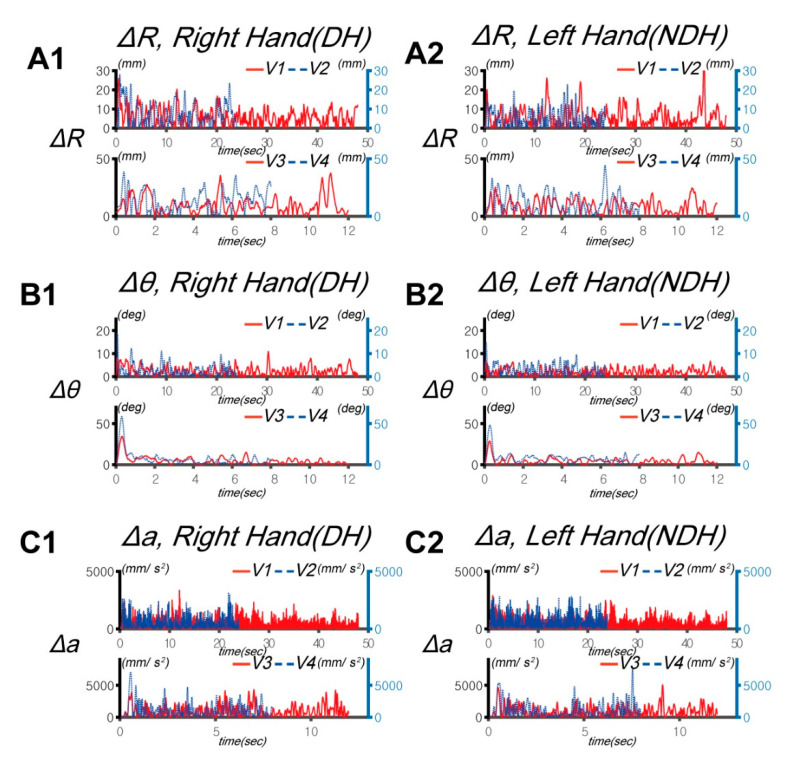
Typical examples of circular tracking movements at 0.125, 0.25, 0.5, and 0.75 Hz (*V*_1_, *V*_2_, *V*_3_, and *V*_4_, respectively). The trajectories of the circular tracking movements of the DH (Dominant Hand) on the frontal plane are indicated for the right hand (CW: **A1**, **B1**, and **C1**) and NDH (Non Dominant Hand) for the left hand (CCW: **A2**, **B2**, and **C2**): Δ*R* for the right hand (**A1**) and left hand (**A2**), Δ*θ* for the right hand (**B1**) and left hand (**B2**), and Δ*a* for the right hand (**C1**) and left hand (**C2**).

**Figure 3 sensors-20-06368-f003:**
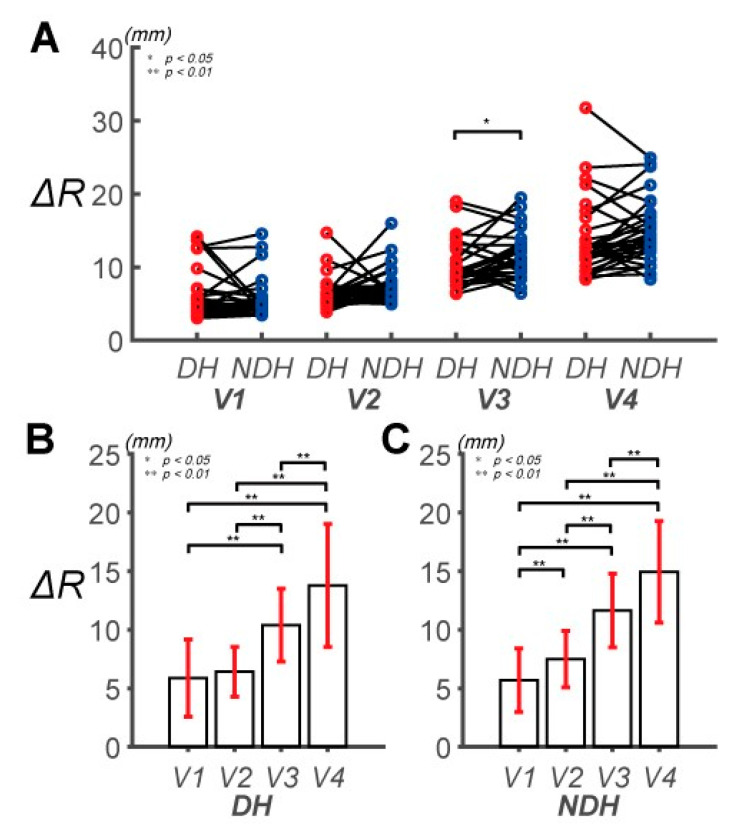
Pairwise comparison of the absolute mean radial position error (Δ*R*) between the DH and NDH. (**A**) Pairwise comparisons of Δ*R* between the subjects’ DHs and NDHs. The speeds of the target are denoted as *V*_1_, *V*_2_, *V*_3_, and *V*_4_. (**B**) Pairwise comparisons of Δ*R* across speeds for the DH. Δ*R* is 5.88 ± 3.30, 6.42 ± 2.12, 10.38 ± 3.11, and 13.78 ± 5.26 mm at *V*_1_, *V*_2_, *V*_3_, and *V*_4_, respectively. (**C**) Pairwise comparisons of Δ*R* across speeds for the NDH. Δ*R* is 5.69 ± 2.72, 7.49 ± 2.42, 11.65 ± 3.15, and 14.95 ± 4.33 mm at *V*_1_, *V*_2_, *V*_3_, and *V*_4_, respectively.

**Figure 4 sensors-20-06368-f004:**
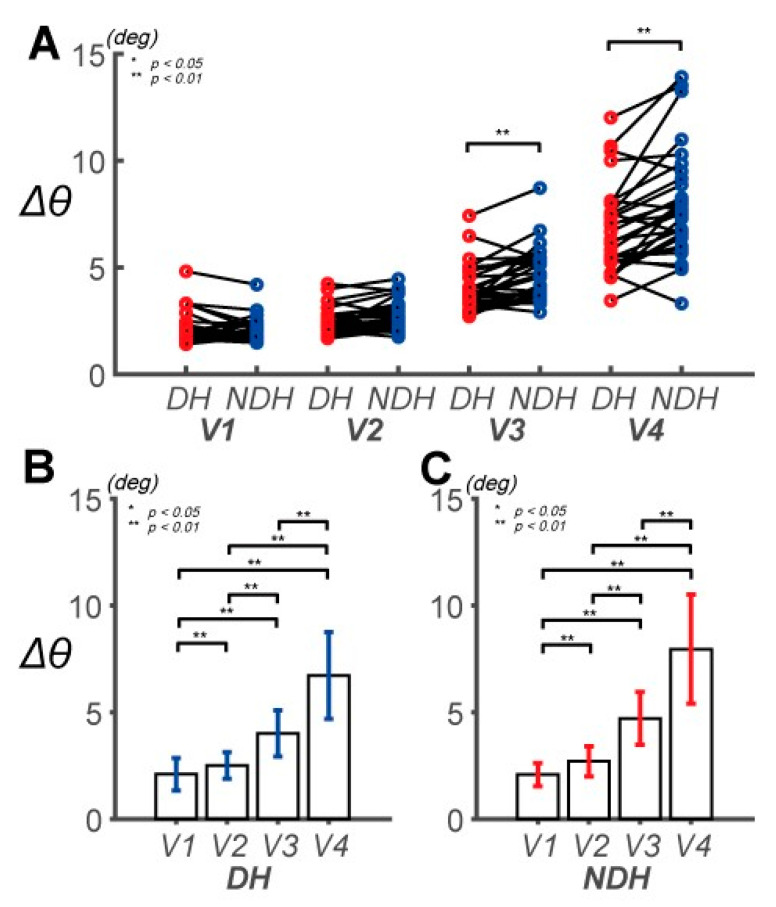
Pairwise comparison of the absolute phase position error (Δ*θ*) between the DH and NDH. (**A**) Pairwise comparisons of Δ*θ* between the subjects’ DHs and NDHs. The speeds of the target are denoted as *V*_1_, *V*_2_, *V*_3_, or *V*_4_. (**B**) Pairwise comparisons of Δ*θ* across speeds for the DH. Δ*θ* is 2.10 ± 0.76, 2.51 ± 0.62, 4.01 ± 1.08, and 6.72 ± 2.04° at *V*_1_, *V*_2_, *V*_3_, and *V*_4_, respectively. (**C**) Pairwise comparisons of Δ*θ* across speeds for the NDH. Δ*θ* is 2.09 ± 0.54, 2.71 ± 0.71, 4.71 ± 1.23, and 7.95 ± 2.55° at *V*_1_, *V*_2_, *V*_3_, and *V*_4_, respectively.

**Figure 5 sensors-20-06368-f005:**
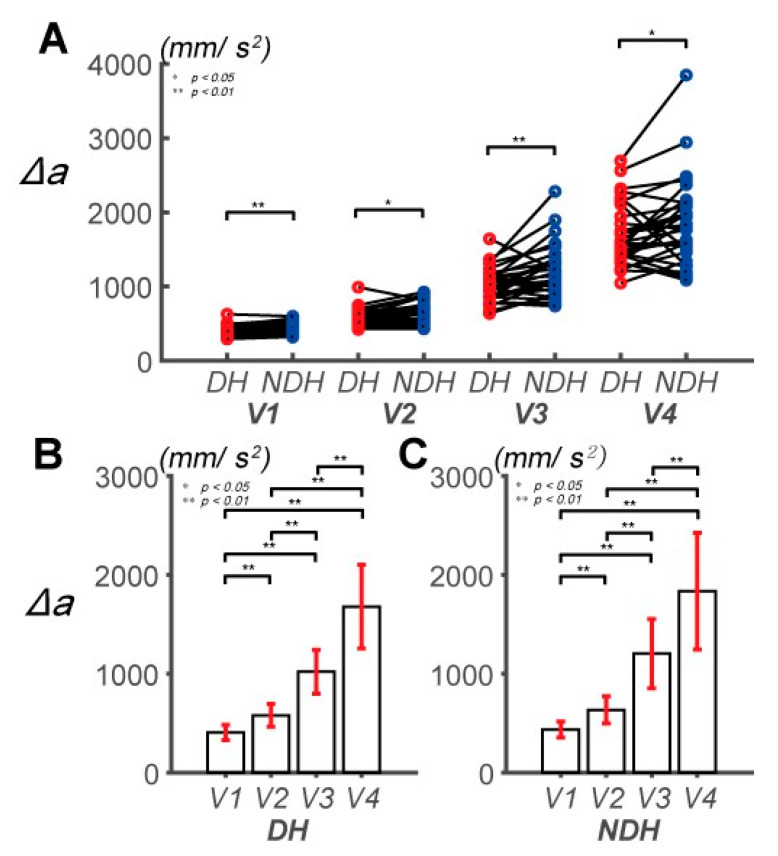
Pairwise comparison of the absolute acceleration error (Δ*a*) between the DH and NDH. (**A**) Pairwise comparisons of Δ*a* between the subjects’ DHs and NDHs. (**B**) Pairwise comparisons of Δ*a* across speeds for the DH. Δ*a* is 406.03 ± 76.14, 578.92 ± 115.52, 1020.77 ± 221.72, and 1679.28 ± 423.38 mm/s^2^, at *V*_1_, *V*_2_, *V*_3_, and *V*_4_, respectively. (**C**) Pairwise comparisons of Δ*a* across speeds for the NDH. Δ*a* is 434.83 ± 80.01, 635.29 ± 138.47, 1203.78 ± 349.08, and 1835.63 ± 589.77 mm/s^2^, at *V*_1_, *V*_2_, *V*_3_, and *V*_4_, respectively.

**Figure 6 sensors-20-06368-f006:**
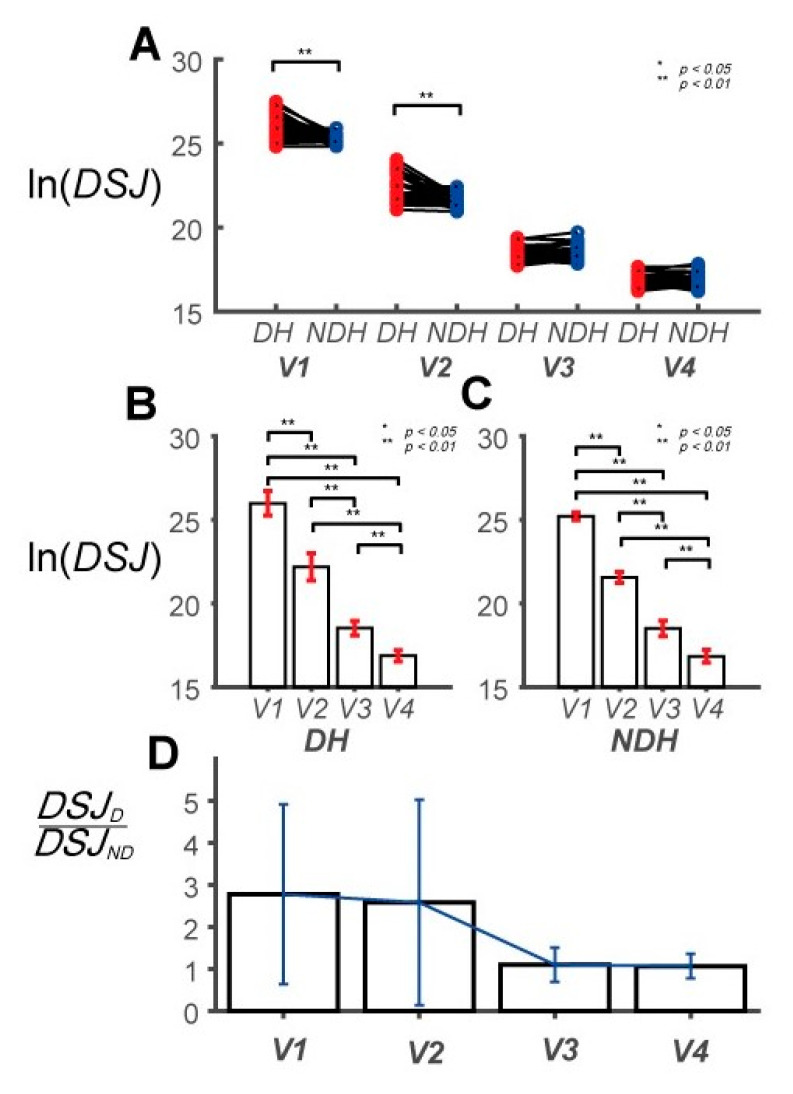
Pairwise comparison of *DSJ* and *DSJ* ratio between the DH and NDH. (**A**) Pairwise comparisons of *DSJ* between the subjects’ DHs and NDHs. The values are calculated using the natural logarithm. (**B**) Pairwise comparisons of *DSJ* across speeds for the *DH*. *DSJ* is 25.97 ± 3.26, 22.19 ± 0.82, 18.53 ± 0.43, and 16.87 ± 0.33, at *V*_1_, *V*_2_, *V*_3_, and *V*_4_, respectively. (**C**) Pairwise comparisons of *DSJ* between the speeds for the *NDH*. *DSJ* is 25.22 ± 0.23, 21.56 ± 0.32, 18.50 ± 0.46, and 16.84 ± 0.39, at *V*_1_, *V*_2_, *V*_3_, and *V*_4_, respectively. (**D**) *DSJ* ratio between the DH and NDH at each speed. The ratios are 2.78 ± 2.14, 2.58 ± 2.44, 1.10 ± 0.41, and 1.07 ± 0.29, at *V*_1_, *V*_2_, *V*_3_, and *V*_4_, respectively.

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
