# Peer review of "Analysis of Visuo Motor Control between Dominant Hand and Non-Dominant Hand for Effective Human-Robot Collaboration"

_sensors, 2020, doi:10.3390/s20216368_

Round 1
Reviewer 1 Report
The paper is potentially very interesting, but its writing needs to be improved. On a conceptual level, I am wondering whether the distinction between DH and NDH is fundamental for the control or whether rather the symmetric vs. antisymmetric aspect is the more basic one, as it emphases the bimanual coordination the impressive effectiveness of which remains unexplained in the present paper. See also Mechsner, F., Kerzel, D., Knoblich, G. et al. Perceptual basis of bimanual coordination. Nature 414, 69–73 (2001) for a more specific case related to this approach (this not my paper, so you don't have to cite it!). In the context of this bimanual representation the characteristics of dominance would also show up, but probably more in terms of time differences or activation profiles. Whether this point of view is more appropriate or the one taken in the paper (as implied by the presented experimental results), would be nicely shown by data that are more intuitive than the traces in Figs 1 and 2, where I am no able identify any characteristics directly from the given data, apart from random fluctuations. Now, it seem that the underlying assuming is that the hands are coded independently, which is counterintuitive and needs therefore more discussion.
Here are a few more detailed comments:
Abstract:
First sentence is not grammatically correct. Should this be "Human-in-the-loop technology requires the research of each hand's etc."?
Introduction:
Many grammatical errors and difficult to read, I would encourage that they get a native English speaker to read through this. E.g. "Human robot collaboration by upper limbs", can be made more clear (human's limbs with a robot or also robot limbs?) do (see also similar expression in the abstract) Should not start second sentence with However, this is a natural extension of the first sentence.
"human-robot collaboration has difficulty [is difficult] due to the different mechanism [mechanisms] between human and robot [used by human and current robots]: robot controls end effector [a robot controls an/the end effector] with position control mechanism based on absolute coordinates, rather human controls [whereas a human/humans controls/control limbs with predictive control mechanisms and feedback control mechanisms based on body coordinates". This is an example with some corrections.
Experimental section:
In Equ. 3, you need to explain what is: v_peak. Also the sentence "DJC has no unit because of its dimensionless characteristics." sounds a bit trivial. My suggestion is to add a better explanation in combination with the explanation of v_peak.Line 151, error in units. s2.
BTW, Instead of DJC also the abbreviation DSJ (Dimensionless Squared Jerk) is used often. please change or add a sentence how DJC is different from DSJ.
It is ultimately an interesting and an worthwhile research report, but it is in need of a rewrite throughout. Introduction and conclusion suddenly bring in "human in the loop technology" into this problem of handedness, and the link should probably be made more clear with some example task or analogy.
Reviewer 2 Report
The manuscript is concerned with investigate differences in visuomotor control between the dominant and non-dominant hands when tracking a target in 3D space. The authors solve current problems and this article is thematically suitable for this journal.
1. The structure of current manuscript is well organized, it is be formulated as follows: Introduction, Motivation, Methodology, Results and discussions, Conclusions.
2. The conclusion of the article is very brief. Please, note about future work and how your work is helpfull for other.
Based on the above, the paper is proposed for minor revision.
I ask that my comments be respected in the revised version of the article.
Reviewer 3 Report
This is an interesting article that addresses the analysis of visuo-motor control comparing the activities between the dominant hand and the non-dominant hand of human beings, using a 3D system in a virtual reality environment. I have some comments about this paper:
- Please, check some typo in the paper, because I have found, on L.132 one, which is “rate55”. It is possible there are some others.
- The axes in the 3D coordinate system, in figure 1, are positioned incorrectly, specifically, axis Z and X are interchanged. Please check this. Therefore, the plots Y vs. X, X vs. Z, and Z vs. Y are not correct. From my point of view, using the right 3D coordinate system, they must be, Y vs. Z, X vs. Z, and Y vs. X, respectively. On the other hand, using the right 3D coordinate system, the three last plots, Y, Z, and X vs time are correct. I only tested the graphics of the right-hand motion.
- I have some remarks about the supplementary tables. First, in the manuscript, they are named with S1 to S4, but in the word document, they are numbered by 1 to 4. Second, maybe these tables must be integrated into the paper as an appendix. This must be consulted with the editor. Third, some numeric values and some items described in the paper are not corresponding with the established in the tables. The authors should carefully review the details described in the paper with the presented in the tables.
- I consider the degree of smoothness for the NDH was less than of the DH, regarding fig. 6a and 6d. However, on lines 431-432 the opposite has been written.
- Finally, I would have liked to see some simple human-robot collaboration experiment, since it is possible to intuit it of the title. However, the human-robot collaboration was only described in the last paragraph of the discussion session and conclusion. However, the suggestion of end-effector motion control of the robot arm up to 471 mm/s is interesting.
